# Sensory Resilience based on Synesthesia

## Abstract

Situated cognition depends on accessing environmental state through sensors. Engineering and cost constraints usually lead to limited "pathways" where, for example, a vision sub-system only includes a camera and the software to deal with it. This traditional and rational design style entails any hardware defect on the pathway causes the system to grind to a halt until repair. We propose a "sensoriplexer" as neural component architecture to address this issue, under the common scenario of multiple sensors availability. This component architecture learns to mix and relate pathways, such that an agent facing failure in a sensory sub-system can degrade gracefully and coherently by relying on its other sub-systems. The architecture is inspired by the concept of synesthesia, and relies on statistical coupling between sensor signals. We show the benefit and limitation of the architecture on a simple shape recognition and a more complex emotion recognition scenarios.

## 1 Introduction

Situated agents embody various degrees of "SPA loops", where they Sense-Process-Act repeatedly in their environment. The simplest agent senses only one dimension in the environment. Sensing failure prevents it from processing and acting—basically causing it to halt. More complex agents with multiple sensors can sometimes continue operating despite the loss of some sensors, but they cannot usually perform anymore the functions related to the lost sensors.

Sensor redundancy[1] is a common solution to ensure continuing operations. Flagship projects like space rovers introduce redundant sensors, such as doubling all "hazcam" on NASA's Curiosity or CNSA's Yutu. Yet engineering and cost constraints block this option in many systems. The loss of most non-critical sensors often means the agent enters a "degraded" mode of operations.

Biological agents can compensate to some extent the loss of a sensor, by using the other ones they are endowed with. A blind person can rely on touch to "read" visual cues encoded to Braille, or "hear" a speech signed visually. Impressive technology allows learning to feel sounds on the skin Bach-y Rita (1972), and by extension for deaf people to "listen" to a conversation by translating sound into tactile patterns on their skin Novich & Eagleman (2015). The compensation capability appears to rely on brain mechanisms that relate different sensory inputs of the same object. These brain mechanisms also appear common to all, with different degree of expression Cytowic (2018). For example, we perceive an apple via (mainly) vision, touch and smell. We can relate the vision of an apple to its likely smell or touch. Although the brain mechanisms are not entirely identified at the present time, the effect has been named synesthesia, sometimes ideaesthesia Jürgens & Nikolić (2012); Nikolić (2014).

This article proposes the *sensoriplexer* component architecture to model synesthesia-inspired mechanisms in artificial agents. The sensoriplexer (SP) allows an agent learning relations between its "senses" for exploitation in its downstream activities. After presenting work related to resilience and their limits, we present a formal model of SP and a corresponding implementation[2]. We use the implementation to conduct a series of experiments to demonstrate and evaluate the capabilities of systems including SP. The article ends with a discussion of the results, and future work.

---

[1] Informally, redundancy means here setting multiple sensors on the same environmental object. Two close-by front cameras on a robot are redundant, but a front and rear camera pair is not (they are *complementary*).

[2] Available in the supplementary materials, and on GitLab (private for now)

## 2 RELATED WORK

The problem at hand is of resiliency: A situated agent is resilient if it can continue operating despite a usually pre-defined range of failure modes. The pre-definition covers explicit problems the agent must deal with (*e.g.* critical safety, guaranteeing a car stops), classes of problems (*e.g.* liveness properties, absence of deadlocks), and sometimes specifications of behaviours under partial failures (*e.g.* a car must be steerable and stoppable even if the air conditioner fails). Resiliency is a broad topic, and we frame its meaning for the extent of this article. An important property of resilience framed this way is its *extent*: We expect a resilient agent to continue operating (as autonomously as possible), yet within compliance, such as safety guarantees.

The sensoriplexer addresses resiliency issues in situated agents, using concepts rooted in Neuroscience and Psychology. Other disciplines have inspired alternative approaches, like redundancy: The loss of a sensor triggers a "switch" to a redundant one. Redundancy is a powerful mean to achieve resilience, but it is not always feasible. Aside cost issues incurred by extra sensors, some systems cannot physically accommodate any extra. Typical quadcopter drones with autopilot capabilities (seen here as situated agents) embark a camera, but they often cannot host a second one, whose payload would exceed the lift capabilities. A similar example is on sensors in satellites and space rovers, where weight and available mount space are very constrained: Redundancy is reserved to safety-critical sensors only. In this paper, we do not see SP as a solution to compensate advanced scientific sensors on satellites, but we expect the concept of synesthesia to inspire future sensor designs for such extreme missions.

Programmable sensors offer an alternative grounded in Electronics and Chemistry. Programmable packages like the CMUCam (where the sensor is not directly programmed) and sensor chips embedding artificial neural networks like the IMX500 from Sony provide agent designers with the capability to programmatically address resiliency issues CMU (2020); Sony (2020). The trend of "intelligent sensors" is relevant to the problem of resilience, yet their application remains limited under the present understanding of their capabilities. A vision sensor can adapt to lighting issues by switching frequency, but cannot relate to or leverage other signals.

On a more theoretical note, we relate this work to the thread of research stemmed from the efficient coding hypothesis, proposed by Barlow, and most notably the statistical representation of signals Barlow (1961). The hypothesis refers to sensory pathways—conceptual routes from a sense of the environment, through areas of the brain, to their final destinations in the brain—for modeling the processing of sensory "messages". Later work shows a statistical dependency between some pathways Simoncelli (2003). SP is an adaptation and implementation of these dependencies in artificial agents. These dependencies are concretely modeled as joint distributions over the different pathways, without explicit relation to any architectural candidate area in the brain (we aim at functional equivalence only, at best).

Another related theory is practopoiesis from Nikolić Nikolić (2014). This theory of "creation of actions" models general principles of abstractions, and introduces the notion of "traverse" along sensory pathways. Traverses separate representation realms, from what we qualify as low-level raw signals to higher-level representations of these signals. The higher-level representations introduce new semantics, typically meaningless at lower levels. This theory can loosely be compared with multi-layer convolutional networks in (for example) character recognition, where the first layers compile low-level glyphs with broad semantics, and last layers capture higher-level concepts of characters. Practopoiesis indicates the number of traverses necessary to achieve a level of representation depends on the task at hand. SP in this article models a tentative traverse from individual sensory paths to an abstract representation of objects represented by the incoming signals. We do not claim SP is a primitive traverse. Its purpose is to model and implement joint representations of objects, and, again, it may not be grounded in any biological apparatus.

A final note pertains to the difference with sensor fusion, often critical to the design of systems like self-driving cars Wikipedia (2020). Sensor fusion is closely related to the functions implemented by SP, but they serve different purposes. Sensor fusion aims at decreasing uncertainty about an object, typically its position or some other property. Sensor fusion relies on adding sensors and "fusing" their data into coherent pictures. The addition continues until acceptable uncertainty is achieved for the given target. SP aims at compensating sensor failures by exploiting other sensors, *i.e.* decreasing uncertainty whenever removing sensors.

## 3    A SENSORIPLEXER MODEL

The environment embedding agents is, from an agent perspective, a set of N signals it can influence in M ways: $Env = \langle (E_i)_{i<N}, (I_j)_{j<M} \rangle$, where each $X_i \in E_i$ is a sequence of signal samples. An agent typically access $n < N$ environmental signals $X_i$ to accomplish its tasks. Each $Y_j \in I_j$ is an "influencable aspect" of the environment. An agent influences $m < M$ aspect of the environment. We note $E = (E_i)_{i<N}$.

A situated agent $Ag$ in $Env$ consists of sets of processes $P$, environmental sensors $S$ and environmental actuators $A$: $Ag = \langle P, S, A \rangle$, where $S = (s_i)_{i<N}$ represents the agent sensing environmental signals $E_i$ for each corresponding $s_i$, and a similar relation on influences. For sake of simplifying the notations, we will drop the actuators from the model, as we focus entirely on the sensory part of the agent. Also, we assume in this presentation that all environmental signals are sequences of length 1, again to simplify the notations. The model focuses however exclusively on sequences (an image would simply be a sequence of length one).

Each process, sensor, and actuator is modeled as a mathematical function $f$ over multiple dimensions[3]:
$$\forall f \in Ag, \exists (n,m) \in \mathbb{N}^{*2}, D_1 \subset \mathbb{C}^n, D_2 \subset \mathbb{C}^m, f : D_1 \to D_2$$

Environmental signals perceived by the agent are modeled as random variable vectors $X_i$, with samples in $E_i$ for each signal $i < N$. Actions in the environments are the output of the actuators, also modeled as random variables $Y_j$, with samples in $I_j$ for each influence $j < M$. Samples in each $X$ and $Y$ are vectors whose dimensions depend on the signal. Here too $X$ and $Y$ are reduced to sequences of length 1 for the presentation of the model. The extensive form of $X$ ($Y$ is similar) becomes: $X = (x_{ij})_{1 \le j \le dim(E_i)}$, for a sample from signal $E_i$.

In the remainder of the article, each function $f$ is represented by a matrix $\boldsymbol{M}$ and shift $s$, such that for a signal $X$, $f(X) = MX + s = Y$. An agent functioning represented with this notation becomes a set of proper compositions over $S \times P \times A$. For a single process, sensor and action in each corresponding set, we get $Ag(X) = A \circ P \circ S(X)$, describing an agent SPA step.

### 3.1    SENSORIPLEXER

The sensoriplexer operates on input signals, right after sensing by the agent. Noted $Sx = (Sx_i)_{i<N}$, we define for a given agent $Ag = \langle P, S, A \rangle$: $Sx : S \to S$, with the given properties:

- Transparency: $Sx_i = Id_E$ when signal is non-null. $\forall X \in E_i, S_i(X) \ne 0_{E_i}, Sx_i(S_i(X)) = S_i(X)$

- Stability: $Sx(0_E) = 0_E$; rest in the absence of sensing.

- Compensation: Whenever an input signal is null as in $S_i(X) = 0$, $Sx_i$ compensates using other signals through some delegate function $\Theta : \prod_i S_i \to \prod_i S_i$: $\forall i \le n, Sx_i(0_{E_i}) = \Theta((E_j)_{j \ne i})$

The $\Theta$ delegate function embodies different compensation methods, which could introduce heuristics specific to a given problem. We propose a generic method in the next section.

It is important to note $Sx$ applies to signals *sensed as null* by the agent. This differs from the absence of signals from the environment: $Sx$ activates on any $X$ such that $S_i(X) = 0$ (null sense), even if $X \ne 0$ (non-null environment signal).

Also, the transparency and compensation requirements are opposing forces, leading to the absence of exact solution in this model. Compensation introduces signal couplings added to each "pure" signal, causing losses in transparency. In terms of linear algebra, this can be represented with:

$$Sx(X) = X + \Theta(X) \tag{1}$$

Where pure transparency would require the compensation term to be null. An agent endowed with a sensoriplexer operates then the overall (simplified) function: $Ag(X) = A \circ P \circ Sx \circ S(X)$.

---

[3]Here too, functions here map over sequences. The formulas use vectors to simplify the exposé. The model and implementation have been designed over tensors to process sequences.

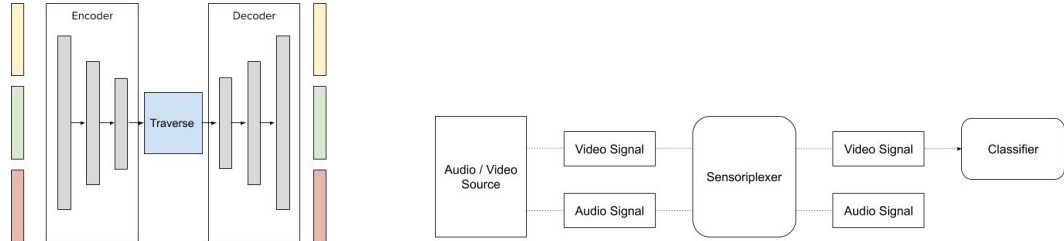

Figure 1: (Left) Sensoriplexer implementation as a stack of auto-encoders with traverse. (Right) Common pattern for all experimental settings.

### 3.2 THE Θ DELEGATE FUNCTION

The delegate function operates on the input signals of the agents, under the properties of the sensoriplexer. We present here an approximate method that complies with all the properties, trying to minimize convolution with the input signals.

We define $\overline{S}_i$ for all $i \leq n$ as $S_i$ complemented with zeros up to $c = \max(\dim(S_j)|j \leq n)$. Doing so all $\overline{S}_i$ have the same dimension $c$. We also define the agent "internal representation" $IR \in \cup_{i \leq n} \overline{S}_i$ as the $nc$-vector (in general, a tensor) obtained by stacking $n$ objects from $\cup_{i \leq n} \overline{S}_i$: $\forall i \leq n, \overline{s}_i \in \overline{S}_i, IR = (\overline{s}_0, \ldots, \overline{s}_n)$. $IR$ represents an object of the environment by a vector gathering vectors from all sensor signals.

We then model the delegate function $\Theta$ with a $nc \times nc$ square matrix, where the $nc$ size serves to apply $\Theta$ to $IR$, and diagonal blocks of size $c$ are all zeros, so $\Theta$ represents the contribution to signal $i$ from all signals except $i$. Together with formula equation 1, SP performs in this model:

$$Sx(IR) = IR + \begin{bmatrix} 0 & B_{12} & \cdots & & B_{1n} \\ B_{21} & 0 & B_{23} & & B_{2n} \\ \vdots & & & & \vdots \\ B_{n1} & \cdots & & B_{n,n-1} & 0 \end{bmatrix} IR \qquad (2)$$

Where $IR = (\overline{s}_0, \ldots, \overline{s}_n)$ is some internal representation vector, and $c \times c$ block matrix $B_{ij}$ is compensation from signal $j$ over $i$. This formulation models SP properties of transparency and stability, and compensates missing senses by adding learned couplings from other signals. The last property of reversibility is expected from an inverse matrix, which results from the auto-encoding overall architecture.

In terms of statistics over the different signals distributions, the SP block matrix formulation is akin to a covariance matrix over $n$ random variables, minus the $nc$ identity matrix. This representation thus captures joint distributions over the $n$ signals integrated by the agent.

We conclude the model presentation with a quick informal example $\Theta$. Let us assume $S = \langle image, sound, smell \rangle$ senses for the agent, each represented by a vector. The $IR_{apple}$ for an apple is vector $IR_{apple} = image + sound + smell$. At some point in time the agent loses the image sense. $IR$ becomes $0 + sound + smell$. $\Theta$ is designed to transform $IR$ into $IR_{apple}$, using the available senses of sound and smell to reproduce a sense of image by minimizing a distance metric (*e.g.* Euclidean distance in our experiments).

### 3.3 OPERATIONAL APPROXIMATION

We present an approach based on Machine Learning to approximate the $\Theta$ delegate function. The approach consists of a training algorithm and a loss function. The model presented in this article requires approximating the block matrices parts of the $\Theta$ delegate function. We perform the approximation by implementing the sensoriplexer as a stack of auto-encoders Hinton & Zemel (1993), with the $\Theta$ function as internal representation.

Figure 1 describes the overall architecture of the SP component. The left coloured boxes are vectors each representing the input layer for a signal, *e.g.* yellow for image, green for sound, and red for temperature. The output of the component on the right corresponds to the input one to one, as indicated by the colouring. The internal layers consist of an encoder collecting the concatenated inputs to feed the encoding layers. The code is then fed into the traverse component, which we model as a square matrix here to implement the block matrix indicated in equation 2 of the Θ delegate. This traverse matrix is initialized as a random matrix. Last, the traverse output is fed into a decoder, symmetric with the encoder, to reconstruct multi-sensory signals. The agent functionalities would then get their inputs from the decoder.

The auto-encoder base implementation of SP complies with the transparency and stability properties. It is an approximation of the SP model, though, as an auto-encoder approximates these properties. The encoder/traverse pair is also a model inspired by the work of Nikolić Nikolić (2014). It encodes raw signal inputs into "higher-level", yet more compact, internal representation reserved to the agent use. In this paper, the agent uses only the decoded representation from the traverse (the traverse code could provide direct clues to the agent—reserved for future work).

## 4 EXPERIMENTS

The purpose of the experiments is to validate both the expected resiliency from adding a sensoriplexer, and the formal model properties, which we expect learned by SP. This paper reports on two complementary experiments: (1) an illustrative extension to a simple shape classifier, (2) an application to emotion recognition.

The shape experiment in (1) aims at demonstrating and quantifying the benefit and cost of introducing SP upstream to a straightforward classifier. The emotion recognition experiment in (2) shows the potential and limitations of SP on a more realistic data set. It also illustrates possible uses as independent module for engineering systems, and stresses the conditions on the simple SP implementation in the frame of this work (as a an auto-encoder with matrix code).

All settings adopt the same organization: We combine an audio-video SP (*i.e.* it accepts as input image and sound frames), with an image-only classifier downstream. We then evaluate for each setting four execution scenarios. Figure 1 pictures the system architecture common to all scenarios. The data source on the left produces streams of images and sound frames, fed into SP. The classifier is plugged to the image output from the sensoriplexer, and produces a class vector as output. Table 1 summarizes the four execution scenarios evaluated for each setting. The loss function is then the sum of the losses for each scenario.

Table 1: Common scenarios evaluated for each setting.

| Scenario Name | Description |
|---|---|
| Direct Signal (DS) | The video signal is directly input to the classifier, effectively bypassing the sensoriplexer. This scenario serves as baseline. |
| Only Video (0V) | Only the video signal is sent to the sensoriplexer. |
| Audio/Video (AV) | Both audio and video signals are sent to the sensoriplexer. |
| Audio Only (A0) | Only the audio signal is sent to the sensoriplexer. |

The Direct Signal scenario serves as baseline, which is the "standard" use of the downstream classifier. The Only Video scenario is same as baseline, going through SP. It allows to evaluate the distortion introduced on the classifier native signal. The Audio/Video scenario adds audio through the SP to evaluate cross-signal distortion. Last, the Audio Only scenario submits audio only to the SP. It shows how much adding the SP allows the classifier becoming resilient to the loss of image. The same classifier without SP would not be applicable to the A0 scenario.

Implicitly, audio and video signals are synchronized and coherent, coming from the same source. We exclude from this evaluation investigating the effect of incoherent or desynchronized output, for this paper to focus first on the potential of the approach.

### 4.1 VISUAL SHAPE CLASSIFICATION FROM SOUND

#### 4.1.1 SETTINGS

We provide a self-contained shape classifier with a synthetic data set, available with the implementation of the sensoriplexer. It consists of a simple classifier trained at identifying 4 shapes in RGB frames. The data set consists of generated 1-second videos, showing the same shape on all frames over a uniform black background, with the spoken word for that shape as audio. Images and voices are synthesized by the implementation. Shapes vary in position, size, colour. Voices vary in speed and pitch, as some examples show on fig. 3 to 6, and 2 (video frames and audio spectrograms).

The code accompanying this paper allows generating datasets of any size. In the reported results, we have used a data set of 1000 audio/video examples, with 85% for training, 5% for validation, and a 10% test holdout. We train the classifier to saturation, as a baseline for comparison.

#### 4.1.2 RESULTS

Table 4.1.2 compiles the classification accuracy results on the test holdout under the four scenarios. Results are based on more than 5 runs with different seeds, with SP configured with $n = 2$ signals, and $c = 32$ vector length for the $IR$ internal representation (the SP core is then a $64 \times 64$ matrix). Each input in the test holdout is submitted four times to the system, once for each scenario. The reported results are therefore consistent and comparable.

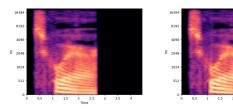

Figure 2: Examples duration and pitch variability in pronouncing "square".

| Scenario | Accuracy ($10^{-2}$) | Impact of SP |
|----------|---------------------|--------------|
| DS | 100% | - |
| 0V | 98.23% | -1.77% (against DS) |
| AV | 97.19% | -2.81% (against DS) |
| A0 | 32.60% | +7.60% (against random) |

Table 2: Shape classification accuracy results.

0V shows the introducing of SP causes an expected performance drop. This simple classifier suffers a drop of about 2%, just by introducing SP. AV shows the addition of the audio signal induces some "noise" inside SP, affecting its outputs. In particular, the video output submitted to the classifier differs more from the original input, and causes a total drop in accuracy of close to 3%. A0 leads to a low performance of 32%, still higher than chance—25% in this 4-class experiment.

Samples in fig. 3-6 show classification results, together with the corresponding output from SP, the actual image input to the classifier. We can observe how A0 achieves reconstructing images from audio, evoking the expected shape. In particular, (a) and (b) show that squares of very different sizes are reconstructed alike under A0, which indicate SP has learned to associate the spoken word "square" to a single, summary representation of the concept. (c) shows a similar example on an ellipse, where the reconstruction from audio in A0 is vague and appears like a rectangle. Last (d) shows the progressive degradation of the input across SP, where a tiny circle is reconstructed with more difficulty when audio is added, illustrating the impact from a multi-signal SP.

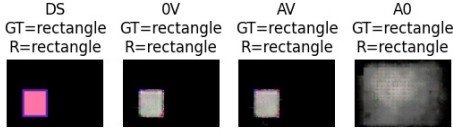

Figure 3: All Successes.



Figure 4: Vagueness in Audio Reconstruction.

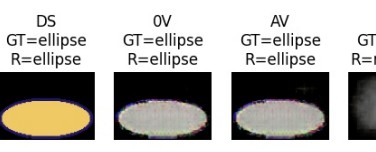

Figure 5: Weak Reconstruction from Audio.



Figure 6: Successive degradation.

The A0 performance shows potential with respect to resiliency. In the absence of SP, the classifier returns whatever output for a null image (or errs due to input shape mismatch). The SP allows the classifier to make some sense out of A0 input signal. A robot endowed with this SP and classifier could continue operating, to some extent, despite the absence of image.

Last, the diagonal blocks of SP's traverse are reported at the end of training with a mean of -0.00, with values ranging from -0.89 to 0.76 over all these blocks. The small mean validates *a posteriori* the traverse model with blocks of zeros on the diagonal (the traverse is initialized with random values).

## 4.2 EMOTION RECOGNITION

### 4.2.1 SETTINGS

This experiment investigates the impact of introducing a sensoriplexer on an arbitrary classifier found on the Internet, as well as a custom classifier created for this paper. We have selected two emotion classifiers found on GitHub—named $Exp_1$ and $Exp_2$ below—with announced respective recognition accuracy of 78% and 74% on the FER 2013 data set (7 emotions) Rajaa (2019); Wu (2018). These two generic classifiers could not generalize well on our selected data set, which differs significantly from their training data. So we also provide results from $Exp_3$, a custom emotion classifier trained on a fraction of our data set (8 emotions).

In this experiment, we train and evaluate the systems on the RAVDESS data set, a set of 8 facial and vocalized expressions acted by professionals Livingstone & Russo (2018). The audio-video files in this data set share a common pattern: A single actor face, centered in the view with a white uniform background, speaking in a silent environment. The files vary in length, how actors express emotions, and frame by frame (the video frames in the shape data set are constant). The data set is then more complex in structure and richness, than the synthetic shape data set in the previous experiment.

Emotion recognition from speech requires at least 200ms of audio Pfister & Robinson (2010). The experiment creates training data by clipping the original files into 300ms-long clips—longer than the minimal requirement—and short enough to meet our computational capabilities. Each example in the training procedure is then a clip and a label (so all clips from the same file share the same label).

The custom classifier $Exp_3$ has been trained on a fraction of RAVDESS itself, using a single video frame every second (videos are 30 frames per second). The purpose of this biased setting is to show the impact of SP on a more realistic data set, not to evaluate the classifier itself. In fact another set of experiments could evaluate the performance of jointly training an SP and downstream systems. These configurations are all valid, and we chose focusing on SP as isolated component in this paper.

### 4.2.2 RESULTS

Table 3 compiles our results on the test holdout under the four scenarios. Results are based on more than 5 runs with different seeds, with SP configured with $n = 2$ signals, and $c = 64$ vector length for $IR$. We do not report here results from runs with $c$ ranging from 2 to 2048: They do show different performance, but did not inform more on SP.

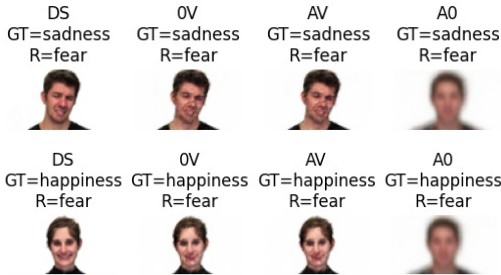

| Scenario | $Exp_1$ | $Exp_2$ | $Exp_3$ |
|----------|---------|---------|---------|
| DS | 16.08% | 13.18% | 52.09 |
| 0V | 13.05% | 13.18% | 36.50 |
| AV | 13.17% | 13.18% | 36.50 |
| A0 | 12.83% | 13.18% | 12.17 |

Table 3: Emotion recognition accuracy results.

Figure 7: (Top) vagueness in audio reconstruction, (bottom) weak reconstruction on difficult images.

The results against $Exp_1$ and $Exp_2$ show a similar trend as with the shape data set. The accuracy with SP becomes here lower than uniformly random choice (for 7 classes, random is about 14.29%).

We have also calculated accuracy using only the central and last frame of each example clip. The performance remains similarly low, so we only report in the table the numbers from the same conditions as the shape experiment, where accuracy is measured on all frames.

We have also confirmed the output from SP were "decent" reproductions of the raw input, as shown in fig. 7 (optimizing regularization Loshchilov & Hutter (2017) and introducing residuals He et al. (2015) did not improve performance significantly). However the DS numbers are the baseline and upper limit performance for the classifiers, which we believe low due to the difference between the original model's data set and RAVDESS. Other scenarios should intuitively be lower, which provides informal validity for our experiments, as well as motivates the next experiment.

$Exp_3$ is simpler than $Exp_1$ and $Exp_2$, but trained on a small subset of RAVDESS exclusively (one video frame per second, less than 0.02% of the data set). This specialization is usually invalid in creating a strong classifier. The purpose here is to see the impact of SP, and opening a perspective on joint training. Table 3 shows the absolute performance is low as a classifier, yet the benefit and cost of SP are coherent with shape classification. Also, the performance in A0 falls at the same level as $Exp_1$ and $Exp_2$. It indicates this SP implementation is unable to reconstruct images well from audio only, which we expect from data significantly more complex than the shapes.

The diagonal blocks of SP's traverse are reported with a 0.00 mean, values ranging from -0.93 to 1.36. Here too the traverse model is validated *a posteriori*, starting from random values. The range of values is also larger than the shape scenario, which we believe related to the more complex data set.

## 5 DISCUSSION & CONCLUSION

The shape classification experiment shows the effect of the sensoriplexer on sensory resilience. The following experiments on emotion classification show potential of the approach in a range of scenarios, despite currently poor results, partially suffering from the limited generalization power of the selected classifiers, and partially for the simple matrix-based model for SP's traverse. The emotion recognition task may lack training data, or require different training features (*e.g.* generating components from the audio cepstrum, also referred to as the MFC mel-frequency components).

SP is also trained independently on the target data set, as an auto-encoder. The experiment with generic emotion classifiers shows a disconnect between SP trained on RAVDESS, and their models trained on FER. Although the two datasets share most of their classes (7 out of 8), the nature of the data varies significantly. RAVDESS provides both higher quality and lower diversity, in its restricted and well-calibrated environment. We expect a joint training of SP, together with its downstream systems to converge to higher overall performance.

SP also provides a mean to build agents resilient to sensory failures. We believe this is useful in itself, yet we expect a future generation of the concept to deal with agent modifications. A typical example is to "upgrade" a sensor component on an active agent. In practice, different sensors have different nominal ranges, even in for the same model (sometimes due to heterogeneous manufacturing practices). SP in this paper could be affected under such change, much as seen in combining a RAVDESS-trained SP and FER-trained classifiers. There is an opportunity for SP to dynamically learn to adapt to a new—yet semantically compatible—sensor.

Synesthesia is reduced in this work to the core meaning of "sensory mixture", which we model as a joint distribution over random variables representing sensory inputs. This decision serves the purpose for an agent to learn how to deal with sensory failures. Synesthesia is however a more elaborated mechanism Cytowic (2018). We expect new insights for artificial agents, from its study in Neurophysiology and Neuroanatomy. Besides, the biological neural correlates are not well understood yet. SP is inspired by synesthesia, rather than an attempt at modeling it. As the understanding of the correlates advances, SP is likely to evolve or disappear, especially if more evidence of a tight coupling between perception and cognition are confirmed.

The sensoriplexer neural component is a practical mechanism for engineering more resilient artificial agents. The experiments in this paper validate the approach under two concrete scenarios, as well as show the potential for applications, such as robotics. The simple architecture selected for presenting SP can be seen as a baseline to explore more capable models, including the introduction of variational or recurrent variants. We will be exploring several of these paths in future work.

ACKNOWLEDGMENTS

The authors would like to thank XXX for reviews on earlier drafts.

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
