# OpenReview forum: "Sensory Resilience based on Synesthesia"
_ICLR.cc/2021/Conference — Reject_

### Official Review · AnonReviewer1 · 2020-10-27
**Major issues with the presentation and significance of empirical results**

**Rating:** 3
**Confidence:** 4

**Review:**

The paper proposes a neural component architecture, named Sensoriplexer (SP), with an aim to introduce resilience to agents performing machine learning tasks. Specifically, if different types of inputs can inform the decision of an agent, the SP aims to learn the dependencies among these inputs to make the agent's decision resilient to the absence of a subset of inputs.

Pros: In theory, the concept of SP is appealing and can have many practical applications. The paper does a decent job motivating the problem and elucidating different aspects of the SP framework with the help of examples.

Cons:
1. Readability and Presentation: Section 3 of the paper, which lays the context for the application of SP, is not clear. The mathematical notations (such as of the form $(E_i)_i, d_E, etc.$) are not adequately defined or discussed. Similarly, the notation $M$ is used to define two separate quantities, which adds to the confusion.

2. Significance of Results: The major issue that I find with the paper is the significance of empirical results. While the notion of adding resilience to ML architectures is the main motivation, the empirical results do not seem to provide convincing evidence for the utility of SP architecture in achieving it. The only result of note in the context seems to be on the visual shape classification task in the first set of experiments, where the 'A0' scenario achieves a better than chance performance (32.60% vs 25%) by the introduction of SP architecture. I am curious if this improvement in performance was enabled by the bias of the ML architecture to classifying one or two specific classes with better accuracy.

The message given by the experiments for the emotion recognition task is unclear to me. The addition of SP architecture does not appear to impact the performance of the three architectures evaluated.

---

> ### Author Response · Authors · 2020-11-16
> **About the interpretation of the empirical results**
>
> Thank you for the summary. We confirm that is what we have attempted with this paper. One minor point: The agent is not restricted to ML tasks. The SP is implemented as such in the frame of this paper, just because auto-encoders allow us approximating the theta function in section 3.2. The agent itself---reduced to its "core function", e.g. a classifier---could well be ResNet, SVM or a closed-form algorithm implementation.
>
> > Cons 1
>
> Section 3 aims at describing the SP model, rather than its context for the applications. We have tried to consider changes to alleviate this situation now, and concluding this understanding may stem from the notations. In fact, we have made an error in typesetting M as a matrix in at least one formula, while also using M as an integer number. The intent was to use ICLR's convention (bold face for matrix), and we missed at least that one place. It seems the M problem is a formatting miss.
>
> A second issue related to typeface is "d_E" in the review. The model contains no such value, but it does use "Id_E" for the identity function over the set E of environmental signals. The "I" does seem "detached" from the "d_E", which is a misuse of Latex (we should be using the identity character).
>
> As for (E_i)i notation, would it be better and sufficient to say once it is the compact representation of (E_1, ..., E_N) ? We detect a misunderstanding on our side, and the problem may be different.
>
> > Cons 2
>
> In the examples we have chosen, A0 is the main focus. An image-based system without SP (e.g. an image classifier) would not be able to leverage anything from an audio signal (encoding mismatch). The SP serves as an adapter to address the encoding mismatch, and a compensation mechanism to leverage the latent shared semantics among signals (image and audio allow us to identify, say, an emotion from the same source, whether signals are used together or independently).
>
> AV is the target of sensor fusion algorithms. In our answer to review 2, we depict the difference with the ViTac system, and we originally tried to distinguish the two in the last paragraph of section 2 of our paper. The SP aims at compensation (there is a part on multi-modal adaptation too), although it does perform fusion in the layers between the inputs and the traverse (the code in the auto-encoding scheme). The rest of SP is actually a form of "defusing" necessary (in this model) for compensating.
>
> We think providing results for AV/0V help understand the cost of introducing SP, while A0 estimates the contribution to the downstream plugged function.

---

### Official Review · AnonReviewer3 · 2020-10-29
**Not novel enough and poor presentation**

**Rating:** 2
**Confidence:** 5

**Review:**

The idea and research direction itself is definitely interesting and worthy of pursuit. However, the execution is really poor. In addition to many improvements to clarity and writing, the proposed method is not at all novel and various variants for reconstructing one sensory modality from others have been proposed in the past:

Gu et al. "Improving domain adaptation translation with domain invariant and specific information" .arXiv [Preprint].arXiv:1904.03879, (2019).

Murez et al.“Image to image translation for domain adaptation,” in Proceedings of the IEEE Conference on Computer Vision and Pattern Recognition (Salt Lake City, UT),4500–4509, (2018).

Luo et al. (2018). “ViTac: feature sharing between vision and tactile sensing for cloth texture recognition,” in2018IEEE International Conference on Robotics and Automation (ICRA) (Brisbane,QLD: IEEE), 2722–2727.

Lee et al. (2019). “Touching to see”and “seeing to feel”: Robotic cross-modal sensory data generation for visual-tactile perception,” in 2019 International Conference on Robotics and Automation (ICRA) (Montreal, QC), 4276–4282.

Tatiya et al. "A Framework for Sensorimotor Cross-Perception and Cross-Behavior Knowledge Transfer for Object Categorization." Frontiers in Robotics and AI 7 (2020): 137.

These papers should be considered by the authors, discussed in related work, and used as a basis to propose something new in future work as there are still problems in that area that need solving.

Finally, the notation is way more complicated than it needs to be for a simple encoder-decoder architecture. My advise to the authors is to simplify it.

---

> ### Author Response · Authors · 2020-11-16
> **On missing related work, novelty and framing**
>
> Thank you for taking the time to list important work. For sake of closure, we review all below. We think there might be novelty in the identification of properties an SP implementation needs to comply with, the a posteriori properties on the trained algorithm, and the specific loss function.
>
> > the notation is way more complicated than it needs to be for a simple encoder-decoder architecture.
>
> SP is a blueprint requiring 3 properties from any implementing method (3.1). Implementing options already exist, including AE, GAN, self-organizing memories and closed-form algorithms. We explain the model by isolating the core mechanism theta in 3.2. This highlights what an implementer needs to do to leverage sensory resilience. We believe the blueprint and this decomposition are the main points, supported experimentally with the matrix of scenarios.
>
> The auto-encoder example is one candidate implementation of theta (3.2). The SP model in section 3 does not require the use of an auto-encoder. The choice is pragmatic for the examples chosen.
>
> Our experiments also apply SP without modification to 4 different tasks from 3 different developers. Most related work may allow similar composability, yet we have included actual results.
>
> > A Framework for Sensorimotor Cross-Perception and Cross-Behavior Knowledge Transfer for Object Categorization
>
> The goals in this article are a superset of ours, and deserves a comparison. It was published on Oct. 9th 2020, and was not available by our submission deadline.
>
> Their paper refers to 2 problems; we share problem (2) on cross-modality. Their paper focuses on a sensorimotor approach. Our work is modeled on a looser notion of data, valid without motor data (e.g. emotion recognition available through a standard browser with access to camera and microphone). We recognize the importance of the sensorimotor approach through the work of Kevin O'Regan. We should cite him to justify and contrast with our choice for synesthesia.
>
> 4.5.1 in their paper reports congruent results with ours in 4 (similar trends and conclusions; their variational version looks better than ours). Our paper, however, proposes a blueprint (sufficient but unlikely necessary) to implement resilience. We believe a contribution is in the model and its evaluation across software from different, unrelated origins.
>
> > Visually Indicated Sounds
>
> The paper presents a RNN to map videos to audio features. Our work is entirely feed-forward, bidirectional map, and valid for other modalities. Also, the paper focuses "on material interaction". We focus on maintaining reliable signals despite sensor failures. Our work relates more to "learn a generically useful representation of the world", compensating signal loss (unlike "cross-modality learning" in Multimodal Deep Learning from Ngiam at al).
>
> > ViTac
>
> The paper makes better texture recognition using visual and tactile data. Our work focuses on recovering from modal failure. In terms of data, ViTac maximises the covariance matrix over all signals. SP also learns a covariance matrix (section 3.2), but calculates the projection of a signal on the hyperplane of all other signals, rather than a covariance optimum.
>
> ViTac:
>
> V -|
>      |fusing --> task using V and T
> T -|
>
> SP:
>
> V -|    |-- V' --> task 1: V only
>      |    |     \
>      |SP|      --> task 2: V and T
>      |    |     /
> T -|    |-- T' --> task 3: T only
>
> SP adds the capability for a task to leverage signals it cannot use by default. For example, task 1 uses a video signal. SP generates video from V, V and T, or T only, transparently for task 1.
>
> An aspect of ViTac we do not automate is "weakly paired situation", when signals have different phases. We simply require getting "enough" samples, common in AV: We rely on "presentation time stamps" in AV codecs, which appears sufficient in our scenarios, but not so for tactile data.
>
> > Image to Image Translation for Domain Adaptation
>
> The paper proposes a solution for domain adaptation. Domain adaptation happens across signals (inter modality) and single signals (intra modality). Our work targets inter modality, whereas their paper focuses on intra modality. Their work identifies properties on learning procedures, while we identify different properties on learning procedures (3.1) and architecture (theta in 3.2). Both approaches share (1) in their method (section 3 on loss definition) and transparency in ours (3.1). Other properties differ: Theirs stem from adversarial settings on images, while ours come from multi-modal compensation.
>
> > Improving Domain Adaptation Translation with Domain Invariant and Specific Information
>
> The paper focuses on translation tasks  to improve domain adaptation by learning both domain specific and agnostic word data. The paper works essentially on word embedding, where our work is likely not applicable. SP embeds raw sensor data into an object space without words. Also, our work aims at cross-modality compensation, rather than domain adaptation.

---

### Official Review · AnonReviewer4 · 2020-10-30
**A model to emulate sensory resilience/fusion using stacked autoencoders. The experiments use audio and video to do shape and emotion classification.**

**Rating:** 5
**Confidence:** 4

**Review:**



+ve Code shared. Power to reproducible science :)

I think a reference to stacked auto-encoders is in order in addition to the reference on auto-encoders. Thank you for citing OG content though! You can also cite a range of papers for redundancy reduction including Bell &Sejnowski, Olshausen & Field

I think the idea is more in line with sensory fusion than synesthesia.

I don’t think I follow what you mean by traverse. It is stated in an obvious fashion but I don’t seem to get the reference.

It would be of value to compare the work against works such as joint audio-video speaker identification/localization. It seems like that would be a fair task to apply against. There are many models for this sort of work out there.

The major concern I have with this work is that while there is a nice narrative on sensory fusion but the model itself is a pretty vanilla stacked auto-encoder with fixed length of inputs. One obvious thing we know about the various sensors in the human system is that their sampling rates are different. For e.g. the time it takes for a photoreceptor to build potential again is different from say a hair cell in the ear. Modeling that perhaps might make the work more actionable.

It could be that I missed some core points in the paper but I don’t get the results in Table 2. Why is a drop in performance an impact of Sensoriplexer and why is this a good thing? Don’t you want to show that you are able to leverage information across modalities and do better, if so should not AV do better?

“Also, the performance in A0 falls at the same level as Exp1and Exp2. It indicates this SP implementation is unable to reconstruct images well from audio only, which we expect from data significantly more complex than the shapes.” — I don’t understand this, I thought the idea was to classifiy emotions.

Overall the paper has some nice ideas but I feel both the models and experiments need a bit more work. Feel free to convince me otherwise :)

---

> ### Author Response · Authors · 2020-11-16
> **Attempt at convincing otherwise, and some more work**
>
> Thank you for the review, and stressing the important point about sensor fusion.
>
> Thank you also for the 2 references about redundancy reduction. We thought the original paper from Barlow, and later work from Simoncelli would cover the topic, and now realize we present them only for the efficient encoding hypothesis, without reference to redundancy reduction. We appreciate the Olshausen and Field reference (which we did not know) for their "overcomplete code" approach provides perspective for future work.
>
> > the idea is more in line with sensory fusion than synesthesia.
>
> The end of section 2 is a paragraph on this specific distinction. SP does perform sensory fusion, but it also regenerates each sense. This last property is necessary to the goal of resilience. In a risky oversimplification, sensor fusion is about "many to one" representation. SP is about "many to many", with each sense "supporting one another". In the review of the ViTac paper in our reply to review 2, we depict the difference between fusion and compensation.
>
> > the various sensors in the human system is that their sampling rates are different.
>
> We deal with different sampling rates in a simple way at this point, by relying on audio-video codecs (a mechanism like "presentation timestamps" to synchronize audio and video). Thisaffects our implementation of the theta function (section 3.2). We believe the model remains valid, but there is definitely an opportunity to contribute more actionable work. We have seen we can extend our work to deal with this problem of "weakly paired" signals (we expand a bit about ViTac in our reply to review 2), yet we have decided to focus on the problem of sensory compensation.
>
> > I don’t think I follow what you mean by traverse.
>
> We use the concept of traverse for the code representing an object from multi-modal data: This code implements the "passage" from raw senses to an object space. For example, an apple looks, smells and tastes like an apple. The senses are clearly distinct, yet the object in object-space is single. The decoder down the traverse aims at extracting each sense from the "object space" code, to mimic what happens in our heads when thinking of an object, e.g. an apple, with evocation from several senses.
>
> > to compare the work against works such as joint audio-video speaker identification/localization.
>
> We left out these fair comparisons as we believe they belong to the sensor fusion category. In these scenarios, audio and video are used jointly to improve on a task. Our settings are for a system to perform the task whenever audio or video becomes unavailable.
>
> > About Table 2.
>
> The drop in performance is expected as SP is a "lossy filter". The properties in section 3.1 create opposing goals in training: Transparency aims at equating input/output of SP (i.e. SP would be identity), whereas compensation forces it to reconstruct a signal from all other signals only (i.e. SP would be a projection, which is not injective). The tension between the 2 desired properties causes the absence of exact solution, thus an approximate solution with training. SP introduces noise in the input signals, so the performance drop. It is not a "good thing" and a limitation of our model.
>
> Several projects use multi-modal approaches to create a better single outcome. They typically use A and V to make a better AV. We aim at a pluggable component that can help a system using only A to leverage V automatically (and vice versa). This is related to the difference between compensation and fusion.
>
> > I thought the idea was to classifiy emotions.
>
> Yes, the example scenario is to classify emotions. The task belongs to the classifier, plugged at the output of SP. The goal of SP is to learn to output as good as possible copies of the input, and the ability to compensate a missing input using all others. The classifier is "unaware" of SP compensating an input, which decouples their functions and allows the composition of SP with other functions.
>
> The function is in practice an off-the-shelf solution, e.g. a ResNet-based emotion classifier. SP is put between the raw input and the ResNet input, with the aim for SP to be (1) transparent (as little noise as possible), (2) if audio is available, be able to use audio input to substitute (or "hallucinate") any missing video input, as such:
>
> Standard setting: V --> ResNet classifier
> Multi-modal SP setting:
> V --|     |--> noisy V --> ResNet Classifier
>       |SP|
> A --|     |--> noisy A --> basically unused (no downstream)
>
> SP aims at producing "noisy V" here, whether V only, AV, or A only is available, and transparently for the downstream classifier.
>
> > +ve Code shared. Power to reproducible science :)
>
> The code submitted with the paper is complete, but there is a bug in the shape dataset generation script (if we remember well on Linux only). Whatever happens to the paper, the code is already open source and the link will be here later (to comply with the anonymous review).

---

### Decision · Program_Chairs · 2021-01-07
**Final Decision**

**Decision:**

Reject

**Comment:**

All three reviewers recommend rejection, based on multiple (mostly shared) concerns. While the authors address the concerns in their rebuttal, the unanimously negative scores remain. I don't see basis to accept the paper.